# IMPROVING SENTENCE REPRESENTATIONS WITH MULTI-VIEW FRAMEWORKS

## ABSTRACT

Multi-view learning can provide self-supervision when different views are available of the same data. Distributional hypothesis provides another form of useful self-supervision from adjacent sentences which are plentiful in large unlabelled corpora. Motivated by the asymmetry in the two hemispheres of the human brain as well as the observation that different learning architectures tend to emphasise different aspects of sentence meaning, we present two multi-view frameworks for learning sentence representations in an unsupervised fashion. One framework uses a generative objective and the other a discriminative one. In both frameworks, the final representation is an ensemble of two views, in which, one view encodes the input sentence with a Recurrent Neural Network (RNN), and the other view encodes it with a simple linear model. We show that, after learning, the vectors produced by our multi-view frameworks provide improved representations over their single-view learnt counterparts, and the combination of different views gives representational improvement over each view and demonstrates solid transferability on standard downstream tasks.

## 1 INTRODUCTION

Multi-view learning methods provide the ability to extract information from different views of the data and enable self-supervised learning of useful features for future prediction when annotated data is not available (de Sa, 1993). Minimising the disagreement among multiple views helps the model to learn rich feature representations of the data and, also after learning, the ensemble of the feature vectors from multiple views can provide an even stronger generalisation ability.

Distributional hypothesis (Harris, 1954) noted that words that occur in similar contexts tend to have similar meaning (Turney & Pantel, 2010), and distributional similarity (Firth, 1957) consolidated this idea by stating that the meaning of a word can be determined by the company it has. The hypothesis has been widely used in machine learning community to learn vector representations of human languages. Models built upon distributional similarity don't explicitly require human-annotated training data; the supervision comes from the semantic continuity of the language data.

Large quantities of annotated data are usually hard and costly to obtain, thus it is important to study unsupervised and self-supervised learning. Our goal is to propose learning algorithms built upon the ideas of multi-view learning and distributional hypothesis to learn from unlabelled data. We draw inspiration from the lateralisation and asymmetry in information processing of the two hemispheres of the human brain where, for most adults, sequential processing dominates the left hemisphere, and the right hemisphere has a focus on parallel processing (Bryden, 2012), but both hemispheres have been shown to have roles in literal and non-literal language comprehension (Coulson et al., 2005; Coulson & van Petten, 2007).

Our proposed multi-view frameworks aim to leverage the functionality of both RNN-based models, which have been widely applied in sentiment analysis tasks (Yang et al., 2016), and the linear/log-linear models, which have excelled at capturing attributional similarities of words and sentences (Arora et al., 2016; 2017; Hill et al., 2016; Turney & Pantel, 2010) for learning sentence representations. Previous work on unsupervised sentence representation learning based on distributional hypothesis can be roughly categorised into two types:

**Generative objective:** These models generally follow the encoder-decoder structure. The encoder learns to produce a vector representation for the current input, and the decoder learns to generate sentences in the adjacent context given the produced vector (Kiros et al., 2015; Hill et al., 2016; Gan et al., 2017; Tang et al., 2018). The idea is straightforward, yet its scalability for very large corpora is hindered by the slow decoding process that dominates training time, and also the decoder in each model is discarded after learning as the quality of generated sequences is not the main concern, which is a waste of parameters and learning effort.

Our first multi-view framework has a generative objective and uses an RNN as the encoder and an invertible linear projection as the decoder. The training time is drastically reduced as the decoder is simple, and the decoder is also utilised after learning. A regularisation is applied on the linear decoder to enforce invertibility, so that after learning, the inverse of the decoder can be applied as a linear encoder in addition to the RNN encoder.

**Discriminative Objective:** In these models, a classifier is learnt on top of the encoders to distinguish adjacent sentences from those that are not (Li & Hovy, 2014; Jernite et al., 2017; Nie et al., 2017; Logeswaran & Lee, 2018); these models make a prediction using a predefined differentiable similarity function on the representations of the input sentence pairs or triplets.

Our second multi-view framework has a discriminative objective and uses an RNN encoder and a linear encoder; it learns to maximise agreement among adjacent sentences. Compared to earlier work on multi-view learning (de Sa, 1993; Dhillon et al., 2011; Wang et al., 2015) that takes data from various sources or splits data into disjoint populations, our framework processes the exact **same** data in two **distinctive** ways. The two distinctive information processing views tend to encode different aspects of an input sentence; forcing agreement/alignment between these views encourages each view to be a better representation, and is beneficial to the future use of the learnt representations.

Our contribution is threefold:

• Two multi-view frameworks for learning sentence representations are proposed, in which one framework uses a generative objective and the other one adopts a discriminative objective. Two encoding functions, an RNN and a linear model, are learnt in both frameworks.

• The results show that in both frameworks, aligning representations from two views gives improved performance of each individual view on all evaluation tasks compared to their single-view trained counterparts, and furthermore ensures that the ensemble of two views provides even better results than each improved view alone.

• Models trained under our proposed frameworks achieve good performance on the unsupervised tasks, and overall outperform existing unsupervised learning models, and armed with various pooling functions, they also show solid results on supervised tasks, which are either comparable to or better than those of the best unsupervised transfer model.

It is shown (Hill et al., 2016) that the consistency between supervised and unsupervised evaluation tasks is much lower than that within either supervised or unsupervised evaluation tasks alone and that a model that performs well on supervised evaluation tasks may fail on unsupervised tasks. It is subsequently showed (Conneau et al., 2017; Subramanian et al., 2018) that, with large-scale labelled training corpora, the resulting representations of the sentences from the trained model excel in both supervised and unsupervised tasks, while the labelling process is costly. Our model is able to achieve good results on both groups of tasks **without** labelled information.

## 2 MODEL ARCHITECTURE

Our goal is to marry **RNN-based** sentence encoder and the **avg-on-word-vectors** sentence encoder into multi-view frameworks with simple objectives. The motivation for the idea is that, RNN-based encoders process the sentences sequentially, and are able to capture complex syntactic interactions, while the avg-on-word-vectors encoder has been shown to be good at capturing the coarse meaning of a sentence which could be useful for finding paradigmatic parallels (Turney & Pantel, 2010).

We present two multi-view frameworks, each of which learns two different sentence encoders; after learning, the vectors produced from two encoders of the same input sentence are used to compose the sentence representation. The details of our learning frameworks are described as follows:

## 2.1 ENCODERS

In our multi-view frameworks, we first introduce two encoders that, after learning, can be used to build sentence representations. One encoder is a bi-directional Gated Recurrent Unit (Chung et al., 2014) $f(s; \phi)$, where $s$ is the input sentence and $\phi$ is the parameter vector in the GRU. During learning, only hidden state at the last time step is sent to the next stage in learning. The other encoder is a linear avg-on-word-vectors model $g(s; \boldsymbol{W})$, which basically transforms word vectors in a sentence by a learnable weight matrix $\boldsymbol{W}$ and outputs an averaged vector.

## 2.2 GENERATIVE OBJECTIVE

Given the finding (Tang et al., 2018) that neither an autoregressive nor an RNN decoder is necessary for learning sentence representations that excel on downstream tasks, our learning framework only learns to predict words in the next sentence. The framework has an RNN encoder $f$, and a linear decoder $h$. Given an input sentence $s_i$, the encoder produces a vector $\boldsymbol{z}_i^f = f(s_i; \phi)$, and the decoder $h$ projects the vector to $\boldsymbol{x}_i = h(\boldsymbol{z}_i^f; \boldsymbol{U}) = \boldsymbol{U} \boldsymbol{z}_i^f$, which has the same dimension as the word vectors $\boldsymbol{v}_w$. Negative sampling is applied to calculate the likelihood of generating the $j$-th word in the $(i+1)$-th sentence, shown in Eq. 1.

$$\log P(w_j | s_i) = \log \sigma(\boldsymbol{x}_i^\top \boldsymbol{v}_{w_j}) + \sum_{k=1}^{K} \mathbb{E}_{w_k \sim P_e(w)} \log \sigma(-\boldsymbol{x}_i^\top \boldsymbol{v}_{w_k}) \tag{1}$$

where $\boldsymbol{v}_{w_k}$ are pretrained word vectors for $w_k$, the empirical distribution $P_e(w)$ is the unigram distribution raised to power 0.75 (Mikolov et al., 2013), and $K$ is the number of negative samples. The learning objective is to maximise the likelihood for words in all sentences in the training corpus.

Ideally, the inverse of $h$ should be easy to compute so that during testing we can set $g = h^{-1}$. As $h$ is a linear projection, the simplest situation is when $\boldsymbol{U}$ is an orthogonal matrix and its inverse is equal to its transpose. Often, as the dimensionality of vector $\boldsymbol{z}_i^f$ doesn't necessarily need to match that of word vectors $\boldsymbol{v}_w$, $\boldsymbol{U}$ is not a square matrix[1]. To enforce invertibility on $\boldsymbol{U}$, a row-wise orthonormal regularisation on $\boldsymbol{U}$ is applied during training, which leads to $\boldsymbol{U}\boldsymbol{U}^\top = \boldsymbol{I}$, where $\boldsymbol{I}$ is the identity matrix, thus the inverse function is simply $h^{-1}(\boldsymbol{x}) = \boldsymbol{U}^\top \boldsymbol{x}$, which is easily computed. The regularisation formula is $||\boldsymbol{U}\boldsymbol{U}^\top - \boldsymbol{I}||_F$, where $|| \cdot ||_F$ is the Frobenius norm. Specifically, the update rule (Cissé et al., 2017) for the regularisation is:

$$\boldsymbol{U} := (1 + \beta)\boldsymbol{U} - \beta(\boldsymbol{U}\boldsymbol{U}^\top)\boldsymbol{U} \tag{2}$$

where $\beta$ is set to 0.01. After learning, we set $\boldsymbol{W} = \boldsymbol{U}^\top$, then the inverse of the decoder $h$ becomes the encoder $g$. Compared to prior work with generative objective, our framework reuses the decoding function rather than ignoring it for building sentence representations after learning, thus information encoded in the decoder is also utilised.

## 2.3 DISCRIMINATIVE OBJECTIVE

Our multi-view framework with discriminative objective learns to maximise the agreement between the representations of a sentence pair across two views if one sentence in the pair is in the neighbourhood of the other one. An RNN encoder $f(s; \phi)$ and a linear avg-on-word-vectors $g(s; \boldsymbol{W})$ produce a vector representation $\boldsymbol{z}_i^f$ and $\boldsymbol{z}_i^g$ for $i$-th sentence respectively. The agreement between two views of a sentence pair $(s_i, s_j)$ is defined as $a_{ij} = a_{ji} = \cos(\boldsymbol{z}_i^f, \boldsymbol{z}_j^g) + \cos(\boldsymbol{z}_i^g, \boldsymbol{z}_j^f)$. The training objective is to minimise the loss function:

$$\mathcal{L}(\phi, \boldsymbol{W}) = - \sum_{|i-j| \leq c} \log p_{ij}, \quad \text{where} \quad p_{ij} = \frac{e^{a_{ij}/\tau}}{\sum_{n=i-N/2}^{i+N/2-1} e^{a_{in}/\tau}} \tag{3}$$

where $\tau$ is the trainable temperature term, which is essential for exaggerating the difference between adjacent sentences and those that are not. The neighbourhood/context window $c$, and the batch size $N$ are hyperparameters.

---

[1] As often the dimension of sentence vectors are equal to or large than that of word vectors, $\boldsymbol{U}$ has more columns than rows. If it is not the case, then regulariser becomes $||\boldsymbol{U}^\top \boldsymbol{U} - \boldsymbol{I}||_F$.

The choice of cosine similarity based loss is based on the observations (Turney & Pantel, 2010) that, of word vectors derived from distributional similarity, vector length tends to correlate with frequency of words, thus angular distance captures more important meaning-related information. Also, since our model is unsupervised/self-supervised, whatever similarity there is between neighbouring sentences is what is learnt as important for meaning.

## 2.4 POSTPROCESSING

The postprocessing step (Arora et al., 2017), which removes the top principal component of a batch of representations, is applied on produced representations from $f$ and $g$ respectively after learning with a final $l_2$ normalisation.

In addition, in our multi-view framework with discriminative objective, in order to reduce the discrepancy between training and testing, the top principal component is estimated by the **power iteration** method (Mises & Pollaczek-Geiringer, 1929) and removed during learning.

## 3 EXPERIMENTAL DESIGN

Three unlabelled corpora from different genres are used in our experiments, including BookCorpus (Zhu et al., 2015), UMBC News (Han et al., 2013) and Amazon Book Review[2](McAuley et al., 2015); six models are trained separately on each of three corpora with each of two objectives. The summary statistics of the three corpora can be found in Table 1. Adam optimiser (Kingma & Ba, 2014) and gradient clipping (Pascanu et al., 2013) are applied for stable training. Pretrained word vectors, fastText (Bojanowski et al., 2017), are used in our frameworks and fixed during learning.

Table 1: **Summary statistics** of the three corpora used in our experiments. For simplicity, the three corpora will be referred to as **1**, **2** and **3** in the following tables respectively.

| Name | # of sentences | mean # of words per sentence |
|---|---|---|
| BookCorpus (**1**) | 74M | 13 |
| UMBC News (**2**) | 134.5M | 25 |
| Amazon Book Review (**3**) | 150.8M | 19 |

Table 2: **Representation pooling in testing phase.** "max($\cdot$)", "mean($\cdot$)", and "min($\cdot$)" refer to global max-, mean-, and min-pooling over time, which result in a single vector. The table also presents the diversity of the way that a single sentence representation can be calculated. $\boldsymbol{X}_i$ refers to word vectors in $i$-th sentence, and $\boldsymbol{H}_i$ refers to hidden states at all time steps produced by $f$.

| Phase | Testing | |
|---|---|---|
| | Supervised | Unsupervised |
| Bi-GRU $f$: $\boldsymbol{z}_i^f$ | $[\max(\boldsymbol{H}_i); \text{mean}(\boldsymbol{H}_i); \min(\boldsymbol{H}_i); \mathbf{h}_i^{M_i}]$ | $\text{mean}(\boldsymbol{H}_i)$ |
| Linear $g$: $\boldsymbol{z}_i^g$ | $[\max(\boldsymbol{W}\boldsymbol{X}_i); \text{mean}(\boldsymbol{W}\boldsymbol{X}_i); \min(\boldsymbol{W}\boldsymbol{X}_i)]$ | $\text{mean}(\boldsymbol{W}\boldsymbol{X}_i)$ |
| Ensemble | Concatenation | Averaging |

All of our experiments including training and testing are done in PyTorch (Paszke et al., 2017). The modified SentEval (Conneau & Kiela, 2018) package with the step that removes the first principal component is used to evaluate our models on the downstream tasks. Hyperparameters, including negative samples $K$ in the framework with generative objective, context window $c$ in the one with discriminative objective, are tuned only on the averaged performance on STS14 of the model trained on the BookCorpus; STS14/G1 and STS14/D1 results are thus marked with a $\star$ in Table 3 and Table 4 to indicate possible overfitting on that dataset/model only. Batch size $N$ and dimension $d$ in both frameworks are set to be the same for fair comparison. Hyperparameters are summarised in supplementary material.

---

[2]Largest subset of Amazon Review.

Table 3: **Results on unsupervised evaluation tasks** (Pearson's $r \times 100$) . **Bold** numbers are the best results among unsupervised transfer models, and underlined numbers are the best ones among all models. 'G' and 'D' refer to generative and discriminative objective respectively. 'WR' refers to the post-processing step that removes the top principal component.

| | Un. Training | | | | | | | | Semi. | | Su. | |
|---|---|---|---|---|---|---|---|---|---|---|---|---|
| | Multi-view | | | | | | fastText | | [2]PSL | | [4]Infer | [3]ParaNMT |
| Task | G1 | G2 | G3 | D1 | D2 | D3 | [12]avg | [1]WR | [1]avg | [1]WR | Sent | (concat.) |
| [6]STS12 | 60.0 | 61.3 | 60.1 | 60.9 | **64.0** | 60.7 | 58.3 | 58.8 | 52.8 | 59.5 | 58.2 | 67.7 |
| [7]STS13 | 60.5 | **61.8** | 60.2 | 60.1 | 61.7 | 59.9 | 51.0 | 59.9 | 46.4 | 61.8 | 48.5 | 62.8 |
| [8]STS14 | 71.1* | 72.1 | 71.5 | 71.5* | **73.7** | 70.7 | 65.2 | 69.4 | 59.5 | 73.5 | 67.1 | 76.9 |
| [9]STS15 | 75.7 | 76.9 | 75.5 | 76.4 | **77.2** | 76.5 | 67.7 | 74.2 | 60.0 | 76.3 | 71.1 | 79.8 |
| [10]STS16 | 75.4 | 76.1 | 75.1 | 75.8 | **76.7** | 74.8 | 64.3 | 72.4 | - | - | 71.2 | 76.8 |
| [11]SICK14 | 73.8 | 73.6 | 72.7 | 74.7 | **74.9** | 72.8 | 69.8 | 72.3 | 66.4 | 72.9 | 73.4 | - |
| Average | 69.4 | 70.3 | 69.2 | 69.9 | **71.4** | 69.2 | 62.7 | 67.8 | - | - | 64.9 | - |

[1]Arora et al. (2017);[2]Wieting et al. (2015);[3]Wieting & Gimpel (2018);[4]Conneau et al. (2017);[5]Wieting & Gimpel (2018); [6−10]Agirre et al. (2012; 2013; 2014; 2015; 2016);[11]Marelli et al. (2014);[12]Mikolov et al. (2017)

Table 4: Comparison with FastSent and QT on STS14 (Pearson's $r \times 100$).

| FastSent (Hill et al., 2016) | | QT (Logeswaran & Lee, 2018) | | Multi-view | | | | | |
|---|---|---|---|---|---|---|---|---|---|
| | +AE | RNN | BOW | G1 | G2 | G3 | D1 | D2 | D3 |
| 61.2 | 59.5 | 49.0 | 65.0 | 71.1* | 72.1 | 71.5 | 71.5* | **73.7** | 70.7 |

## 3.1 Unsupervised Evaluation - Textual Similarity Tasks

**Representation:** For a given sentence input $s$ with $M$ words, suggested by (Pennington et al., 2014; Levy et al., 2015), the representation is calculated as $\boldsymbol{z} = \left(\hat{\boldsymbol{z}}^f + \hat{\boldsymbol{z}}^g\right)/2$, where $\hat{\boldsymbol{z}}$ refers to the post-processed and normalised vector, and is mentioned in Table 2.

**Tasks**: The unsupervised tasks include five tasks from SemEval Semantic Textual Similarity (STS) in 2012-2016 (Agirre et al., 2012; 2013; 2014; 2015; 2016) and the SemEval2014 Semantic Relatedness task (SICK-R) (Marelli et al., 2014).

**Comparison**: We compare our models with: • *Unsupervised learning*: We selected models with strong results from related work, including fastText, fastText+WR. • *Semi-supervised learning*: The word vectors are pretrained on each task (Wieting et al., 2015) without label information, and word vectors are averaged to serve as the vector representation for a given sentence (Arora et al., 2017). • *Supervised learning*: ParaNMT (Wieting & Gimpel, 2018) is included as a supervised learning method as the data collection requires a neural machine translation system trained in supervised fashion. The InferSent[3] (Conneau et al., 2017) trained on SNLI (Bowman et al., 2015) and MultiNLI (Williams et al., 2017) is included as well.

The results are presented in Table 3. Since the performance of FastSent (Hill et al., 2016) and QT (Logeswaran & Lee, 2018) were only evaluated on STS14, we compare to their results in Table 4.

All six models trained with our learning frameworks outperform other unsupervised and semi-supervised learning methods, and the model trained on the UMBC News Corpus with discriminative objective gives the best performance likely because the STS tasks contain multiple news- and headlines-related datasets which is well matched by the domain of the UMBC News Corpus.

## 3.2 Supervised Evaluation

The evaluation on these tasks involves learning a linear model on top of the learnt sentence representations produced by the model. Since a linear model is capable of selecting the most relevant dimensions in the feature vectors to make predictions, it is preferred to concatenate various types of representations to form a richer, and possibly more redundant feature vector, which allows the

---

[3]The released InferSent (Conneau et al., 2017) model is evaluated with the postprocessing step.

Table 5: **Supervised evaluation tasks. Bold** numbers are the best results among unsupervised transfer models, and underlined numbers are the best ones among all models. "†" refers to an ensemble of two models. "‡" indicates that additional labelled discourse information is required. Our models perform similarly or better than existing methods, but with higher training efficiency.

| Model | Hrs | SICK-R | SICK-E | MRPC | TREC | MR | CR | SUBJ | MPQA | SST |
|---|---|---|---|---|---|---|---|---|---|---|
| [1]Conneau et al. (2017);[2]Arora et al. (2017);[3]Hill et al. (2016); [4]Kiros et al. (2015); [5]Gan et al. (2017);[6]Jernite et al. (2017);[7]Nie et al. (2017);[8]Tai et al. (2015);[9]Zhao et al. (2015) [10]Le & Mikolov (2014);[11]Logeswaran & Lee (2018);[12]Shen et al. (2018);[13]Mikolov et al. (2017) | | | | | | | | | | |
| *Supervised task-dependent training - No transfer learning* | | | | | | | | | | |
| [9]AdaSent | - | - | - | - | 92.4 | 83.1 | 86.3 | 95.5 | 93.3 | - |
| [8]TF-KLD | - | - | - | 80.4/85.9 | - | - | - | - | - | - |
| [12]SWEM-*concat* | - | - | - | 71.5/81.3 | 91.8 | 78.2 | - | 93.0 | - | 84.3 |
| *Supervised training - Transfer learning* | | | | | | | | | | |
| [1]InferSent | <24 | 88.4 | 86.3 | 76.2/83.1 | 88.2 | 81.1 | 86.3 | 92.4 | 90.2 | 84.6 |
| *Unsupervised training with unordered sentences* | | | | | | | | | | |
| [10]ParagraphVec | 4 | - | - | 72.9/81.1 | 59.4 | 60.2 | 66.9 | 76.3 | 70.7 | - |
| [2]GloVe+WR | - | 86.0 | 84.6 | - / - | - | - | - | - | - | 82.2 |
| [13]fastText+bow | - | - | - | 73.4/81.6 | 84.0 | 78.2 | 81.1 | 92.5 | 87.8 | 82.0 |
| [3]SDAE | 72 | - | - | 73.7/80.7 | 78.4 | 74.6 | 78.0 | 90.8 | 86.9 | - |
| *Unsupervised training with ordered sentences* | | | | | | | | | | |
| [3]FastSent | 2 | - | - | 72.2/80.3 | 76.8 | 70.8 | 78.4 | 88.7 | 80.6 | - |
| [4]Skip-thought | 336 | 85.8 | 82.3 | 73.0/82.0 | 92.2 | 76.5 | 80.1 | 93.6 | 87.1 | 82.0 |
| [5]CNN-LSTM † | - | 86.2 | - | 76.5/83.8 | 92.6 | 77.8 | 82.1 | 93.6 | 89.4 | - |
| [6]DiscSent ‡ | 8 | - | - | 75.0/ - | 87.2 | - | - | 93.0 | - | - |
| [7]DisSent ‡ | - | 79.1 | 80.3 | - / - | 84.6 | 82.5 | 80.2 | 92.4 | 89.6 | 82.9 |
| [11]MC-QT | 11 | 86.8 | - | 76.9/**84.0** | **92.8** | 80.4 | 85.2 | 93.9 | 89.4 | - |
| **Multi-view** G1 | 3.5 | **88.1** | 85.2 | 76.5/83.7 | 90.0 | 81.3 | 83.5 | 94.6 | 89.5 | 85.9 |
| **Multi-view** G2 | 9 | 87.8 | **85.9** | **77.5**/83.8 | 92.2 | 81.3 | 83.4 | 94.7 | 89.5 | 85.9 |
| **Multi-view** G3 | 9 | 87.7 | 84.4 | 76.0/83.7 | 90.6 | 84.0 | 85.6 | 95.3 | 89.7 | 88.7 |
| **Multi-view** D1 | 3 | 87.9 | 84.8 | 77.1/83.4 | 91.8 | 81.6 | 83.9 | 94.5 | 89.1 | 85.8 |
| **Multi-view** D2 | 8.5 | 87.8 | 85.2 | 76.8/83.9 | 91.6 | 81.5 | 82.9 | 94.7 | 89.3 | 84.9 |
| **Multi-view** D3 | 8 | 87.7 | 85.2 | 75.7/82.5 | 89.8 | 85.0 | 85.7 | 95.7 | 90.0 | 89.6 |

linear model to explore the combination of different aspects of encoding functions to provide better results.

**Representation:** Inspired by prior work (McCann et al., 2017; Shen et al., 2018), the representation $z^f$ is calculated by concatenating the outputs from the global mean-, max- and min-pooling on top of the hidden states $H$, and the last hidden state, and $z^g$ is calculated with three pooling functions as well. The post-processing and the normalisation step is applied individually. These two representations are concatenated to form a final sentence representation. Table 2 presents the details.

**Tasks**: Semantic relatedness (SICK) (Marelli et al., 2014), paraphrase detection (MRPC) (Dolan et al., 2004), question-type classification (TREC) (Li & Roth, 2002), movie review sentiment (MR) (Pang & Lee, 2005), Stanford Sentiment Treebank (SST) (Socher et al., 2013), customer product reviews (CR) (Hu & Liu, 2004), subjectivity/objectivity classification (SUBJ) (Pang & Lee, 2004), opinion polarity (MPQA) (Wiebe et al., 2005). The results are presented in Table 5.

**Comparison**: Our results as well as related results of supervised task-dependent training models, supervised learning models, and unsupervised learning models are presented in Table 5. Note that, for fair comparison, we collect the results of the best single model of MC-QT (Logeswaran & Lee, 2018) trained on BookCorpus.

Six models trained with our learning frameworks either outperform other existing methods, or achieve similar results on some tasks. The model trained on the Amazon Book Review gives the best performance on sentiment analysis tasks, since the corpus conveys strong sentiment information.

Table 6: **Ablation study on our multi-view frameworks**. Variants of our frameworks are tested to illustrate the advantage of our multi-view learning frameworks. In general, under the proposed frameworks, learning to align representations from both views helps each view to perform better and an ensemble of both views provides stronger results than each of them. The arrow and value pair indicate how a result differs from our multi-view learning framework. Better view in colour.

| UMBC News | Hrs | Unsupervised tasks | Supervised tasks | | MRPC |
|---|---|---|---|---|---|
| | | Avg of STS tasks (STS12-16, SICK14) | Avg of SICK-R, STS-B | Avg of Binary-CLS tasks (MR, CR, SUBJ, MPQA, SST) | |
| **Our Multi-view with Generative Objective + Invertible Constraint** | | | | | |
| $z^f$ | | 66.6 | 82.0 | 86.1 | 74.7/83.1 |
| $z^g$ | 9 | 67.8 | 82.3 | 85.3 | 74.8/82.2 |
| $en(z^f, z^g)$ | | 70.3 | 82.7 | 87.0 | 77.5/83.8 |
| Generative Objective without Invertible Constraint | | | | | |
| $z^f$ | | 55.7 (↓10.9) | 79.9 (↓2.1) | 86.0 (↓0.1) | 73.2/81.7 |
| $z^g$ | 9 | 70.1 (↑2.3) | 82.8 (↑0.5) | 85.0 (↓0.3) | 74.3/82.0 |
| $en(z^f, z^g)$ | | 67.8 (↓2.5) | 82.9 (↑0.2) | 86.4 (↓0.7) | 74.8/83.2 |
| **Our Multi-view with Discriminative Objective**: $a_{ij} = \cos(z_i^f, z_j^g) + \cos(z_i^g, z_j^f)$ | | | | | |
| $z^f$ | | 67.4 | 83.0 | 86.6 | 75.5/82.7 |
| $z^g$ | 8 | 69.2 | 82.6 | 85.2 | 74.3/82.7 |
| $en(z^f, z^g)$ | | 71.4 | 83.0 | 86.6 | 76.8/83.9 |
| Multi-view with $f_1$ and $f_2$: $a_{ij} = \cos(z_i^{f_1}, z_j^{f_2}) + \cos(z_i^{f_2}, z_j^{f_1})$ | | | | | |
| Multi-view with $g_1$ and $g_2$: $a_{ij} = \cos(z_i^{g_1}, z_j^{g_2}) + \cos(z_i^{g_2}, z_j^{g_1})$ | | | | | |
| $z^{f_1}$ | 17 | 49.7 (↓17.7) | 82.2 (↓0.8) | 86.3 (↓0.3) | 75.9/83.0 |
| $en(z^{f_1}, z^{f_2})$ | | 57.3 (↓14.1) | 81.9 (↓1.1) | 87.1 (↑0.5) | 77.2/83.7 |
| $z^{g_1}$ | 2 | 68.5 (↓0.7) | 80.8 (↓1.8) | 84.2 (↓1.0) | 72.5/82.0 |
| $en(z^{g_1}, z^{g_2})$ | | 69.1 (↓2.3) | 77.0 (↓6.0) | 84.5 (↓2.1) | 73.5/82.3 |
| $en(z^{f_1}, z^{g_1})$ | 19 | 67.5 (↓3.9) | 82.3 (↓0.7) | 86.9 (↑0.3) | 76.6/83.8 |
| Single-view with $f$ only: $a_{ij} = \cos(z_i^f, z_j^f)$, Single-view with $g$ only: $a_{ij} = \cos(z_i^g, z_j^g)$ | | | | | |
| $z^f$ | 9 | 57.8 (↓9.6) | 81.6 (↓1.4) | 85.8 (↓0.8) | 74.8/82.3 |
| $z^g$ | 1.5 | 68.7 (↓0.5) | 81.1 (↓1.5) | 83.3 (↓1.9) | 72.9/81.0 |
| $en(z^f, z^g)$ | 10.5 | 68.6 (↓2.8) | 82.3 (↓0.7) | 86.3 (↓0.3) | 75.4/82.5 |

# 4 DISCUSSION

In both frameworks, RNN encoder and linear encoder perform well on all tasks, and generative objective and discriminative objective give similar performance.

## 4.1 GENERATIVE OBJECTIVE: REGULARISATION ON INVERTIBILITY

The orthonormal regularisation applied on the linear decoder to enforce invertibility in our multi-view framework encourages the vector representations produced by $f$ and those by $h^{-1}$, which is $g$ in testing, to agree/align with each other. A direct comparison is to train our multi-view framework without the invertible constraint, and still directly use $U^\top$ as an additional encoder in testing. The results of our framework with and without the invertible constraint are presented in Table 6.

The ensemble method of two views, $f$ and $g$, on unsupervised evaluation tasks (STS12-16 and SICK14) is averaging, which benefits from aligning representations from $f$ and $g$ by applying invertible constraint, and the RNN encoder $f$ gets improved on unsupervised tasks by learning to align with $g$. On supervised evaluation tasks, as the ensemble method is concatenation and a linear model is applied on top of the concatenated representations, as long as the encoders in two views process sentences distinctively, the linear classifier is capable of picking relevant feature dimensions from both views to make good predictions, thus there is no significant difference between our multi-view framework with and without invertible constraint.

### 4.2 DISCRIMINATIVE OBJECTIVE: MULTI-VIEW VS. SINGLE-VIEW

In order to determine if the multi-view framework with two different views/encoding functions is helping the learning, we compare our framework with discriminative objective to other reasonable variants, including the multi-view model with two functions of the same type but parametrised independently, either two $f$-s or two $g$-s, and the single-view model with only one $f$ or $g$. Table 6 presents the results of the models trained on UMBC News Corpus.

As specifically emphasised in previous work (Hill et al., 2016), linear/log-linear models, which include $g$ in our model, produce better representations for unsupervised evaluation tasks than RNN-based models do. This can also be observed in Table 6 as well, where $g$ consistently provides better results on unsupervised tasks than $f$. In addition, as expected, multi-view learning with $f$ and $g$, improves the resulting performance of $f$ on unsupervised tasks, also improves the resulting $g$ on supervised evaluation tasks.

Provided the results of models with generative and discriminative objective in Table 6, we confidently show that, **in our multi-view frameworks with $f$ and $g$, the two encoding functions improve each other's view.**

### 4.3 ENSEMBLE IN MULTI-VIEW FRAMEWORKS

**In general, aligning the representations generated from two distinct encoding functions ensures that the ensemble of them performs better.** The two encoding functions $f$ and $g$ encode the input sentence with emphasis on different aspects, and the subsequently trained linear model for each of the supervised downstream tasks benefits from this diversity leading to better predictions. However, on unsupervised evaluation tasks, simply averaging representations from two views without aligning them during learning leads to poor performance and it is worse than $g$ (linear) encoding function solely. Our multi-view frameworks ensure that the ensemble of two views provides better performance on both supervised and unsupervised evaluation tasks.

Compared with the ensemble of two multi-view models, each with two encoding functions of the same type, our multi-view framework with $f$ and $g$ provides slightly better results on unsupervised tasks, and similar results on supervised evaluation tasks, while our model has much higher training efficiency. Compared with the ensemble of two single-view models, each with only one encoding function, the matching between $f$ and $g$ in our multi-view model produces better results.

## 5 CONCLUSION

We proposed multi-view sentence representation learning frameworks with generative and discriminative objectives; each framework combines an RNN-based encoder and an average-on-word-vectors linear encoder and can be efficiently trained within a few hours on a large unlabelled corpus. The experiments were conducted on three large unlabelled corpora, and meaningful comparisons were made to demonstrate the generalisation ability and transferability of our learning frameworks and consolidate our claim. The produced sentence representations outperform existing unsupervised transfer methods on unsupervised evaluation tasks, and match the performance of the best unsupervised model on supervised evaluation tasks.

Our experimental results support the finding (Hill et al., 2016) that linear/log-linear models ($g$ in our frameworks) tend to work better on the unsupervised tasks, while RNN-based models ($f$ in our frameworks) generally perform better on the supervised tasks. As presented in our experiments, multi-view learning helps align $f$ and $g$ to produce better individual representations than when they are learned separately. In addition, the ensemble of both views leveraged the advantages of both, and provides rich semantic information of the input sentence. Future work should explore the impact of having various encoding architectures and learning under the multi-view framework.

Our multi-view learning frameworks were inspired by the asymmetric information processing in the two hemispheres of the human brain, in which the left hemisphere is thought to emphasise sequential processing and the right one more parallel processing (Bryden, 2012). Our experimental results raise an intriguing hypothesis about how these two types of information processing may complementarily help learning.

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

# APPENDIX

## 1 EVALUATION TASKS

The details, including size of each dataset and number of classes, about the evaluation tasks are presented below[1].

Table 1: Details about the evaluation tasks used in our experiments.

| Task name | |Train| | |Test| | Task | Classes |
|---|---|---|---|---|
| **Relatively Small-scale** | | | | |
| MR | 11k | 11k | sentiment (movies) | 2 |
| CR | 4k | 4k | product reviews | 2 |
| SUBJ | 10k | 10k | subjectivity/objectivity | 2 |
| MPQA | 11k | 11k | opinion polarity | 2 |
| TREC | 6k | 0.5k | question-type | 6 |
| SICK-R | 4.5k | 4.9k | semantic textual similarity | 6 |
| STS-B | 5.7k | 1.4k | semantic textual similarity | 6 |
| MRPC | 4k | 1.7k | paraphrase | 2 |
| SICK-E | 4.5k | 4.9k | NLI | 3 |
| **Relatively Large-scale** | | | | |
| SST-2 | 67k | 1.8k | sentiment (movies) | 2 |

## 2 POWER ITERATION

The Power Iteration was proposed in Mises & Pollaczek-Geiringer (1929), and it is an efficient algorithm for estimating the top eigenvector of a given covariance matrix. Here, it is used to estimate the top principal component from the representations produced from $f$ and $g$ separately. We omit the superscription here, since the same step is applied to both $f$ and $g$.

Suppose there is a batch of representations $\mathbf{Z} = [\mathbf{z}_1, \mathbf{z}_2 ..., \mathbf{z}_N] \in \mathbb{R}^{2d \times N}$ from either $f$ or $g$, the Power Iteration method is applied here to estimate the top eigenvector of the covariance matrix[2]: $\mathbf{C} = \mathbf{Z}\mathbf{Z}^\top$, and it is described in Algorithm 1:

---
**Algorithm 1** Estimating the First Principal Component (Mises & Pollaczek-Geiringer, 1929)
---
    **Input:** Covariance matrix $\mathbf{C} \in \mathbb{R}^{2d \times 2d}$, number of iterations $T$
    **Output:** First principal component $\mathbf{u} \in \mathbb{R}^{2d}$
1:  Initialise a unit length vector $\mathbf{u} \in \mathbb{R}^{2d}$
2: **for** $t \leftarrow 1, T$ **do**
3:     $\mathbf{u} \leftarrow \mathbf{C}\mathbf{u}$,
4:     $\mathbf{u} \leftarrow \dfrac{\mathbf{u}}{||\mathbf{u}||}$

---

In our experiments, $T$ is set to be 5.

---
[1] Provided by https://github.com/facebookresearch/SentEval
[2] In practice, often $N$ is less than $2d$, thus we estimate the top eigenvector of $\mathbf{Z}^\top\mathbf{Z} \in \mathbb{R}^{N \times N}$.

## 3    TRAINING & MODEL DETAILS

The hyperparameters we need to tune include the batch size $N$, the dimension of the GRU encoder $d$, and the context window $c$, and the number of negative samples $K$. The results we presented in this paper is based on the model trained with $N = 512$, $d = 1024$. Specifically, in discriminative objective, the context window is set $c = 3$, and in generative objective, the number of negative samples is set $K = 5$. It takes up to 8GB on a GTX 1080Ti GPU.

The initial learning rate is $5 \times 10^{-4}$, and we didn't anneal the learning rate through the training. All weights in the model are initialised using the method proposed in He et al. (2015), and all gates in the bi-GRU are initialised to 1, and all biases in the single-layer neural network are zeroed before training. The word vectors are fixed to be those in the FastText (Bojanowski et al., 2017), and we don't finetune them. Words that are not in the FastText's vocabulary are fixed to 0 vectors through training. The temperature term is initialised as 1, and is tuned by the gradient descent during training.

The temperature term is used to convert the agreement $a_{ij}$ to a probability distribution $p_{ij}$ in Eq. 1 in the main paper. In our experiments, $\tau$ is a trainable parameter initialised to 1 that decreased consistently through training. Another model trained with fixed $\tau$ set to the final value performed similarly.

## 4    EFFECT OF POST-PROCESSING STEP

Table 2: **Effect of the Post-processing Step**. 'WR' refers to the post-processing step (Arora et al., 2017) which removes the principal component of a set of learnt vectors. The postprocessing step overall improves the performance of our models on unsupervised evaluation tasks, and also improves the models with generative objective on supervised sentence similarity tasks. However, it doesn't have a significant impact on single sentence classification tasks.

| Model | WR | Unsupervised tasks | Supervised tasks | | |
|---|---|---|---|---|---|
| | | Avg of STS tasks (STS12-16, SICK14) | Avg of SICK-R, STS-B | Avg of Binary-CLS tasks (MR, CR, SUBJ, MPQA, SST) | MRPC |
| **Our Multi-view with Generative Objective + Invertible Constraint** | | | | | |
| G1 | w/ | 69.4 | 83.1 | 87.0 | 76.5/83.7 |
| G1 | w/o | 66.5 | 80.0 | 86.6 | 76.7/83.5 |
| G2 | w/ | 70.3 | 82.7 | 87.0 | 77.5/83.8 |
| G2 | w/o | 67.7 | 79.5 | 86.3 | 78.3/84.6 |
| G3 | w/ | 69.2 | 83.2 | 88.6 | 76.0/83.7 |
| G3 | w/o | 65.0 | 80.0 | 88.6 | 76.1/83.7 |
| **Our Multi-view with Discriminative Objective**: $a_{ij} = \cos(\boldsymbol{z}_i^f, \boldsymbol{z}_j^g) + \cos(\boldsymbol{z}_i^g, \boldsymbol{z}_j^f)$ | | | | | |
| D1 | w/ | 69.9 | 82.6 | 87.0 | 77.1/83.4 |
| D1 | w/o | 59.3 | 81.8 | 84.2 | 75.0/82.5 |
| D2 | w/ | 71.4 | 83.0 | 86.6 | 76.8/83.9 |
| D2 | w/o | 68.5 | 83.5 | 86.5 | 76.5/84.3 |
| D3 | w/ | 69.2 | 83.2 | 89.1 | 75.7/82.5 |
| D3 | w/o | 64.0 | 83.4 | 89.2 | 75.1/82.8 |

## 5 COMBINING BOTH GENERATIVE AND DISCRIMINATIVE OBJECTIVE IN OUR MULTI-VIEW FRAMEWORK

Models with both generative and discriminative objectives are trained to see if further improvement can be provided by combining an RNN encoder, an inverse of a linear decoder in the generative objective and a linear encoder in the discriminative objective. The results of models trained on BookCorpus and UMBC News are presented in Table 3.

As presented in the table, no further improvement against models with only one objective is shown. In our understanding, the inverse of the linear decoder in generative objective behaves similarly to the linear encoder in the discriminative objective, which is presented in Table 6 in the main paper. Therefore, combining two objectives doesn't perform better than only one of them.

Table 3: **Our multi-view framework with both generative and discriminative objective**. 'GD1' refers to a model with both generative and discriminative objectives trained on BookCorpus. The results here don't show significant difference against the model trained with only one objective.

| | | Unsupervised tasks | Supervised tasks | | |
| | | Avg of STS tasks | Avg of | Avg of Binary-CLS tasks | |
| **Model** | Hrs | (STS12-16, SICK14) | SICK-R, STS-B | (MR, CR, SUBJ, MPQA, SST) | MRPC |
| --- | --- | --- | --- | --- | --- |
| G1 | 3.5 | 69.4 | 83.1 | 87.0 | 76.5/83.7 |
| D1 | 3 | 69.9 | 82.6 | 87.0 | 77.1/83.4 |
| GD1 | 4 | 68.0 | 82.6 | 87.1 | 76.4/83.8 |
| G2 | 9 | 70.3 | 82.7 | 87.0 | 77.5/83.8 |
| D2 | 8.5 | 71.4 | 83.0 | 86.6 | 76.8/83.9 |
| GD2 | 10 | 70.5 | 83.1 | 87.1 | 76.5/84.0 |

## 6 NUMBER OF PARAMETERS

The number of parameters of each of the selected models is:

1. Ours: $\approx 8.8M$
2. Quick-thought (Logeswaran & Lee, 2018): $\approx 19.8M$
3. Skip-thought (Kiros et al., 2015): $\approx 57.7M$

