# OpenReview forum: "Improving Sentence Representations with Multi-view Frameworks"
_ICLR.cc/2019/Conference_

### Official Review · AnonReviewer3 · 2018-11-02
**This paper presents a multiview framework for sentence representation in NLP tasks. But novelty appears too limited here.**

**Rating:** 5
**Confidence:** 4

**Review:**

This paper presents a multiview framework for sentence representation in NLP tasks. Authors propose two architectures, one using a generative objective, while the other uses a discriminative objective. Both combine a recurrent based encoding function and a linear model. Large experiments have been conducted on several NLP tasks and datasets, showing improvement of the introduced frameworks compared to baselines.

The paper is globally well written and has a clear presentation. But I'm not sure to understand why authors motivate their work on the asymmetric information processing in the two hemispheres of the human brain. It sounds like a nice motivation, but the work presented here does not show any clear answer for this, except the idea of combining two different encoders for sentence representation..

My main concern is about the term multiview since the merging step is somewhat trivial (min/max/averaging vectors or concatenation). This is far from significant works on multiview learning, see: "Multi-view learning overview: Recent progress and new challenges".

Table 3, where G and D refer respectively to Generative and Discriminative models. But what differences between G1, G2, G3 ; D1, D2, D3 ?

Invertible constraint is a nice idea for using inverse of the decoder as the encoder. Is it really to take advantages of decoder information on the encoder/representation part? Or also to reduce the amount of parameters learnt in the model? Moreover, it is unclear on the ablation study: did you consider the original encoder ; or still the inverse of decoder but without the constraint? Unfortunately, it seems to not give significant gain, according to ablation study in table 6.

In this current form, I feel this paper does not give sufficient novelty to be accepted at ICLR.

---

> ### Author Response · Authors · 2018-11-10
> **Thanks for your review, but we vehemently disagree that our multi-view method is only merging (min/max/averaging or concatenation).**
>
> Our comparison methods in the ablation study do just merging and don’t perform well. (for instance merging an independently trained two encoding functions: f and g)  We show however that when we force the views to agree (consensus maximisation) (and thus change the learnt representations of each view) we get improved performance of the individual views (and also of their subsequently merged representation).   The main point is by requiring consensus maximisation, we achieve better representations for each view.   The merging after that gives an additional improved performance.
>
> We agree that we have not proved anything related to brain hemispheric processing but are only giving it for motivation.  We will make it clear that we have not proved anything in that regard.
>
> We are using the understanding of the term multi-view as originally envisioned in the introduction to the Proceedings of the ICML 2005 Workshop on Learning with Multiple Views, which states “Contributions to this workshop from fields of machine learning, such diverse as clustering, semi-supervised learning, named entity recognition or ensemble learning show that there is a strong interest in learning problems with multiple represented instances and consensus maximizing learning methods in a variety of communities”. Our method is using doubly represented instances and consensus maximisation and is a novel type of unsupervised multi-view learning.
>
>
> Clarifications:
>
>
> 1/ “G” and “D” refer to generative objective and discriminative objective respectively. “1”, “2” and “3” refer to three corpora used in our experiments, which are Toronto BookCorpus, UMBC News and Amazon Book Review. “G1” refers to the model trained with generative objective on Toronto BookCorpus. (see Table 1, we also mentioned it in Table 2, but we didn’t mention it again in other tables.)
>
>
> 2/ In both generative and discriminative objective, an RNN encoder (f) and a linear encoder (g) are trained under our multi-view frameworks, and also both applied during testing. Table 6 states comparisons between our multi-view frameworks and other variants.
>
> Specifically, in models with generative objective, the RNN encoder (f) is still used after learning, and its performance is in rows indicated by z^f.
>
>
> 3/ Interestingly, the z^g’s are worse in the generative without invertible constraint but the ensembled (from invertible constraint) with z^f(which is usually worse than z^g) is better than the ensembled without invertible constraint.

---

> > ### Author Response · Authors · 2018-12-05
> > **In response to the request for comparison to [1] and also in response to Reviewer 2 regarding the generality of our approach**
> >
> > We note that none of the examples in [1] create new views de novo but rely on natural complementary views (e.g. from audio and video; or from “links to the page” and “words on the page”). Here we are constructing two views in feature space (representing the processing resulting from a complex RNN and a simple linear network) from a single view (exact same data) in input space (the sentences) when only one natural view is available.
> >
> > We believe that this approach could be applied to videos using both the idea that neighbouring video frames should encode to a similar representation AND that complex (e.g. convolutional neural networks) and simple encoding function (e.g. SIFT Flow [2]) should provide similar representations and could help each view to learn better representations by maximising their consensus. To the best of our knowledge, this has not been tried (with the Minimising-Disagreement/Consensus Maximisation training objective on videos alone).
> >
> > If however, this approach is not useful for unsupervised video representation learning, then our method reveals something deep about these standard encoding functions (the standard recurrent networks and linear methods used in the field) for sentence representations.
> >
> > [1] Wang, Weiran et al. “On Deep Multi-View Representation Learning.” ICML (2015).
> > [2] Liu, Ce et al. “SIFT Flow: Dense Correspondence across Scenes and Its Applications.” IEEE Transactions on Pattern Analysis and Machine Intelligence 33 (2011): 978-994.

---

### Official Review · AnonReviewer2 · 2018-11-02
**contributions for ICLR community are unclear; after revisions and discussion, I am more positive**

**Rating:** 6
**Confidence:** 5

**Review:**

After reading through the authors' comments and rereading parts of the submission, I have become a little more positive about this paper.

I am still unsure about the contributions to the ICLR community. The authors merely state "We believe our contributions to ICLR community are clear and valuable" without backing up this claim with an argument.

But in the rest of the authors' comment, they make some good points. I think those points should be made more prominently in the paper itself. I would suggest that the authors describe their approach as using different, complementary encoders of the input sentence and consensus maximization. If they wish to describe this as multi-view learning, that's fine, but I think using the term "consensus maximization" (or something more descriptive like that) in prominent places would be helpful.

If the approach is applicable beyond sentence embedding learning, then it would behoove the authors to describe the approach in a general way so that readers will see how they can apply it to their own tasks. As currently written, the paper is very much focused on sentence embedding learning, which causes me to think that the paper is more appropriate for an NLP venue. But it is true that ICLR publishes papers that are application-specific, so I can't consider this to be a deal-breaker for the paper.

I raised my score to a 6.

--------------------------------- original review follows: ----------------------------------

This paper describes experiments in learning sentence embeddings from unlabeled text. The paper compares a few different compositional architectures and training objectives. The story of the paper focuses on the training of multiple architectures jointly for a single sentence, then ensembling those architectures at test time to represent sentences. One architecture is an RNN and the other is a word averaging model, and the idea is that these two architectures capture different "views" of the sentence.

Pros:

As a general-purpose method to get sentence embeddings without using any resources other than unannotated text documents, this approach has strong results, including solid performance on the SentEval tasks and relatively-low training times.

It was also nice to see how the results depend on the domain of the training data. Review data definitely helps on the several sentiment-related tasks, which provides further evidence of a worrisome aspect of SentEval.

Cons:

Overall, the paper feels incremental and is likely a better fit for an NLP conference. What are the generalizable contributions to the ICLR community? Given the known differences between RNNs and word averaging models for sentences (especially on the SentEval tasks, which, as the authors note, was discussed by Hill et al.), it's entirely unsurprising that combining the two would be a good idea. But even if this were not the case, the ubiquity of ensembles outperforming single models in deep learning also makes it unsurprising that combining these two kinds of model architectures would be beneficial. So I'm just not sure if there is a significant contribution beyond the NLP results. These results seem solid (though a bit incremental), but if the primary contribution is empirical, then the paper would be a better fit for an NLP venue.

In addition, I'm not sure if "multi-view" is an appropriate description of the approach. In Sec. 1, we find the sentence "Compared to earlier work on multi-view learning (de Sa, 1993; Dhillon et al., 2011) that takes data from various sources or splits data into disjoint populations, our framework processes the exact same input data in two distinctive ways."  Therefore, maybe it's not quite accurate to describe this approach using the term "multi-view learning"? I think it would make more sense to use a different term rather than stretch the definition for a well-known one.

I kept expecting the paper to present results when combining the generative and discriminative objectives, but as far as I can tell, this was never done. What would happen if one were to use multitask learning and just optimize the sum of the two losses?

I'd suggest citing and comparing to the results from "Learning General Purpose Distributed Sentence Representations via Large Scale Multi-task Learning" by Subramanian et al. (ICLR 2018) and the Byte mLSTM from "Learning to generate reviews and discovering sentiment" by Radford et al.

I'm not sure how excited we should get about not using any annotations or structured resources for learning sentence embeddings. The authors do not motivate this goal.


Below are more specific comments/questions:

Sec. 2.2 contains the sentence "Ideally, the inverse of h should be easy to compute so that during testing we can set g = h^-1." At this point in the paper, it is not clear what g is going to be applied to at test time, since presumably the following sentence is not going to be available at test time, right? I think it would be good to discuss how the model is going to be used at test time before discussing the inverse of h.

Sec. 2.3:
In Eq. (3), why does the denominator sum always start at 1 no matter what i and j are? That is, why would the denominator always sum over the first N sentences in the dataset?

I think the pooling methods in Table 2 should be described in Section 2.

In Table 2, it is not clear what h_i^{M_i} is. If M_i is the number of words in sentence i, that should be mentioned somewhere.

Sec. 3.1:
"For a given sentence input s with M words, suggested by (Pennington et al., 2014; Levy et al., 2015), the representation is calculated as z = (\hat{z}_f + \hat{z}_g) / 2, where \hat{z} refers to the post-processed and normalised vector, and is mentioned in Table 2."  I don't understand. Where is this mentioned in Table 2?


Minor issues follow:

Sec. 1:
"Distributional hypothesis" --> "the distributional hypothesis"
"in machine learning community" --> "in the machine learning community"
"and distributional hypothesis" --> "and the distributional hypothesis"
"the linear/log-linear models" --> "linear/log-linear models"
"based on distributional hypothesis" --> "based on the distributional hypothesis"
"contraint" --> "constraint"

Sec. 2:
"marry RNN-based sentence encoder" --> "marry RNN-based sentence encoders" or "marry the RNN-based sentence encoder"

Sec. 2.1:
"only hidden state" --> "only the hidden state"

Sec. 2.2:
"prior work with generative objective" --> "prior work with generative objectives"

Sec. 2.3:
"with discriminative objective learns" --> "with the discriminative objective learns"

Sec. 3.1:
In Table 3, I don't see where superscript 5 is shown in the table.

Sec. 3.2:
In Table 5, the numbered superscripts at the top of the table do not show up next to the methods in the actual table rows.

---

> ### Author Response · Authors · 2018-11-10
> **Thanks for your review and your suggestions. However, we believe our contributions to ICLR community are clear and valuable.**
>
> We believe our contributions to ICLR community are clear and valuable.
>
>
> 1/ Multi-view learning is the appropriate term for our paper.
>
> From the introduction to the Proceedings of the ICML 2005 Workshop on Learning with Multiple Views [1], the 3rd paragraph goes
>
> “Contributions to this workshop from fields of machine learning, such diverse as clustering, semi-supervised learning, named entity recognition or ensemble learning show that there is a strong interest in learning problems with multiply represented instances and consensus maximizing learning methods in a variety of communities”.
>
> Our method is using doubly represented instances and consensus maximization and is a novel type of unsupervised multi-view learning.  Recognizing that different types of networks create different features that can be effectively used for multi-view learning is part of our contribution.
>
>
>
> 2/ We agree with you that combining different representations is a good idea.  However, our contribution goes well beyond that (in fact that is the baseline that we compare to in the ablation studies).  The point of our paper is that training to maximise the agreement between the views
> improves their individual results (with no ensembling used)  and that combining them (after multi-view training) does better than an ensemble from independently trained views (f: an RNN and g: linear).
>
>
>
> 3/ It is not guaranteed that the ensemble of single models outperforms each single one of them; it also depends on the tasks.
>
> The essence of most ensemble learning methods, including boosting, bag-of-trees, random forest is the majority voting mechanism when making predictions, and it has been proved that the performance improvement is guaranteed.
>
> However, in the unsupervised evaluation tasks presented in our paper, the similarity of two sentences are determined by directly measuring the cosine similarity of two sentence representations, and a direct ensemble of different encoding functions won’t necessarily give better performance than each one of them.
>
> In fact, as presented in Table 6, in a model with the generative objective but without invertible constraint (not our multi-view frameworks), the ensemble of two encoding functions performs poorly, and even worse than the linear encoding function solely. Meanwhile, our model with the invertible constraint (where the inverted decoder must be in agreement with the forward encoder)  provides better performance when taking an ensemble of two encoding functions. It is also the case in the models with the discriminative objective. Furthermore, the ensemble representations in our multi-view frameworks (where the two representations are trained to agree) perform better than other variants in general.
>
> Given the results we presented, we think that it is important to encourage consensus from different views. Multi-view frameworks we proposed in our paper could also be applied in other areas, as it is an idea of marrying multiple information processing methods in a single learning algorithm.
>
>
>
> 4/ We will motivate learning sentence representations from unlabelled corpora better in our next version.   We would like to note that we are not using completely unstructured data as the corpora contain temporally meaningful data and our methods critically use this consistency in meaning between neighbouring sentences.
>
> In our paper, we referred to several interesting accepted-to-ICLR papers working on unsupervised sentence representation learning [2][3], and also papers that were published on other venues on the same topic.  As the multi-view learning aspect provides the core novelty of our paper, we focused more on motivating that contribution and the importance of multi-view learning.  We will update our paper to include detailed motivation for the importance of unsupervised sentence representation learning.
>
>
> 5/ It is interesting to try the multi-task learning that combines generative and discriminative objectives, but we think the final performance will be similar to learning with only one of them, given the results of generative and discriminative objective are comparable to each other. We will try it and report results in our later version.
>
>
> Replies to specific questions:
> Sec. 2.2: How to use the decoder at test time is explained later in the same section, and we directly take the transpose of the linear decoder as another encoder at test time.
> Sec. 2.3: Agreed. The summation in the denominator should range from j-N/2 to j+N/2-1
>
>
> [1] Rüping and Scheffer. "Learning with multiple views." Proc. ICML Workshop on Learning with Multiple Views. 2005.
> [2] Arora et al. “A simple but tough-to-beat baseline for sentence embeddings.” In ICLR2017.
> [3] Logeswaran and Lee. “An efficient framework for learning sentence representations.” In ICLR2018.

---

> > ### Author Response · Authors · 2018-12-05
> > **In response to the request for comparison to [1] and also in response to Reviewer 2 regarding the generality of our approach**
> >
> > We note that none of the examples in [1] create new views de novo but rely on natural complementary views (e.g. from audio and video; or from “links to the page” and “words on the page”). Here we are constructing two views in feature space (representing the processing resulting from a complex RNN and a simple linear network) from a single view (exact same data) in input space (the sentences) when only one natural view is available.
> >
> > We believe that this approach could be applied to videos using both the idea that neighbouring video frames should encode to a similar representation AND that complex (e.g. convolutional neural networks) and simple encoding function (e.g. SIFT Flow [2]) should provide similar representations and could help each view to learn better representations by maximising their consensus. To the best of our knowledge, this has not been tried (with the Minimising-Disagreement/Consensus Maximisation training objective on videos alone).
> >
> > If however, this approach is not useful for unsupervised video representation learning, then our method reveals something deep about these standard encoding functions (the standard recurrent networks and linear methods used in the field) for sentence representations.
> >
> > [1] Wang, Weiran et al. “On Deep Multi-View Representation Learning.” ICML (2015).
> > [2] Liu, Ce et al. “SIFT Flow: Dense Correspondence across Scenes and Its Applications.” IEEE Transactions on Pattern Analysis and Machine Intelligence 33 (2011): 978-994.

---

### Official Review · AnonReviewer1 · 2018-11-03
**Impressive results and interesting model**

**Rating:** 7
**Confidence:** 4

**Review:**

This paper is about a multi-view framework for learning sentence representations. Two objective functions (a generative one and a discriminative one) are proposed that make use of two encoders, one of them is based on an RNN and the other on a linear projection of averaged word embeddings. Each of these objective functions has a multi-view framework where their respective objective functions are in part based on making sure their is some relationship between the two different encoders. This multi-view framework is shown to be helpful over having independent encoders in their ablation study.

The authors evaluate on the SentEval benchmark (a collection of tasks where a shallow neural network is learned and the sentence embeddings are kept fixed) and a collection of STS tasks (where the cosine between two vectors is used to estimate their semantic similarity).

The results are impressive. A closer examination of them though leaves me with some questions and thoughts.

Regarding the SentEval numbers, I would like to know the dimensionality of your models in Table 5. I am somewhat unclear of how the final embeddings were produced, it seems that you concatenate mean, max, min, and last_h from the RNN encoder and then mean, max, min from the projection encoder. Is that correct? That would make your feature vector 7*1024 dimensions, which is a bit bigger than most of what you compare to (some of these methods are 4096 dims). With this type of evaluations, larger feature vectors do help performance, thought I am certain that you would have nice performance even if your dimension was reduced (this is from looking at the ablation). I think making the dimensions more explicit and clarifying in the text how the final feature vector was created would be helpful for readers.

Another thing to consider in the evaluation, is that a paper recently pointed out that max pooling in a certain way in SentEval can artificially inflate results for some of these datasets. I noticed that max-pooling is used in your experiments. This paper also shows how big of an effect larger feature vectors have on performance: https://openreview.net/forum?id=BkgPajAcY7. I'd like to know if your results are affected by this max-pooling operation as is the case for several well-known papers in this area.

I also noticed that a lot of the best SentEval numbers came from using the book review dataset. This makes a lot of sense in that a lot of these are based on sentiment and is something that was in part used by (Radford et al. 2017) to obtain strong performance on these tasks. A similar thing happens with the STS data and the news domain as noted in this paper.

I noticed you did the principal component removal trick for InferSent, but it did not have a large effect on performance. How big of an effect did if have with your methods? I'm glad you included it in InferSent, but I'd like to see this as well in the ablation.

Overall I do think this paper has value for the community. It shows how strong results can be obtained using just raw text and using less parameters and training faster than other recent approaches. I do think a lot of the gains here are due to clever design choices in their experiment (for instance using different types of raw data which help more on certain tasks, removing the first principal component, etc.) but putting everything together to get very competitive results with across all these tasks with an interesting approach and an accompanying analysis is a nice contribution.

Minor comment: The paper was tough to understand in parts due to symbols/abbreviations not being defined or motivated clearly. It'd be nice if the authors could define the symbols/abbreviations that are in the tables in the captions. An example of this would be WR in Table 4. The left-most column in Table 6 could also be clearer (I know it is in the text, but I was confused about what f1 and f2 represent etc. in my first pass). This also occurs in the text as well like when g is introduced in Section 2.2.

PROS:
- Interesting and novel model combining RNNs and word-averaging
- I find the multiview framework to be a nice contribution, having the models tied in this way also improves performance.
- Model is fast to train and requires only raw text
- Competitive results with SOA on many datasets - both those requiring a trained classifier using the fixed embedding and STS tasks.

CONS
- Some of their gains are due to choice of dataset for training or removing the first principal component - advantages that other comparable models may or may not have. Not really a con though, more of an observation. I would like to see an ablation to see the effect of removing the first principal component.

---

> ### Author Response · Authors · 2018-11-10
> **Thanks for your detailed review; we really appreciate all the time you put into it.  We will definitely also explain the symbols in the captions of tables.**
>
> 1/ Our paper makes separate contributions.   A significant contribution is that training the views to agree with each other results in better representations for each of the views after training.   This finding is not dependent on the benefits of max pooling or large dimensionality as our comparisons in the ablation tests use the same representation pooling methods and dimensionality.
>
> In Table 6, we compared our proposed method with other variants that don’t maximise the agreement between two views, and the results show that 1) the individual encoders in our method perform better and 2) the ensemble of two views also overall works better than other ensembles. The hyperparameters, including representation pooling and dimensionality of the models, are the same, which indicate that our multi-view learning frameworks help and provide better performance.
>
>
> 2/ Another contribution is that the combination after multi-view training is uniformly better on the unsupervised tasks and on average better on the supervised tasks.   We believe that a good representation should perform well in both types of tasks and our goal was to create the best system for finding a great representation for both types of tasks.
>
>
> 3/ The ICLR submission [3] you mentioned is very interesting, and we think it is definitely important to understand how much gain we can get by learning against an ensemble of randomly initialised models. However, their paper only demonstrates the performance gain from increased model dimensionality on supervised evaluation tasks; no results on unsupervised evaluation tasks are provided.   We have found that larger dimensionality often harms performance when evaluated on unsupervised tasks.  Following the findings of [4], we have also noticed that good performance on supervised tasks is not consistent with good performance on unsupervised tasks.  This observation was a motivating point for our paper.
>
> We are interested in finding a representational system that works for both types of tasks and believe that our simple method of freely (without training a different model) creating a larger dimensionality representation (that works well for supervised tasks) from a relatively lower dimensional model (that works well for unsupervised tasks)  is another generally useful contribution.
>
> Also, all of our studies in the ablation table (Table 6) use similar representation pooling methods and still demonstrate a benefit to multi-view learning.
>
>
> 4/ In our proposed multi-view frameworks, removing the first principal component leads to better performance on all unsupervised evaluation tasks, but not on the supervised evaluation tasks. It has been shown by at least two papers [1][2] that removing first principal component provides stronger performance on unsupervised tasks, and the results show that the assumption of representations being isotropic is helpful.
>
> While on supervised evaluation tasks, additional learning of a linear classifier is required, it is unclear if removing the first principal component would definitely help or not. We will report the results in our next version.
>
>
>
> [1] Arora et al. A simple but tough-to-beat baseline for sentence embeddings. In ICLR2017.
> [2] Mu et al. “All-but-the-Top: Simple and Effective Postprocessing for Word Representations.” In ICLR2018.
> [3] Anonymous, “No Training Required: Exploring Random Encoders for Sentence Classification.” In submission to ICLR2019.
> [4] Hill et al. “Learning Distributed Representations of Sentences from Unlabelled Data.” HLT-NAACL (2016).

---

### Author Response · Authors · 2018-11-18
**Revision 1 to include experiments and ablation study suggested by Reviewer 1 and Reviewer 2**

We first include an ablation study suggested by Reviewer 1, and an additional experiment suggested by Reviewer 2, and they are in the appendix now.



1/ The effect of the post-processing step. (suggested by Reviewer 1)

Six models trained on BookCorpus, UMBC news and Amazon Book Review with generative or discriminative objective are evaluated without the post-processing step that removes the first principal component. The results are presented in Table 2 in the appendix.

Overall, the postprocessing step overall improves the performance of our models on unsupervised evaluation tasks, and also improves the models with generative objective on supervised sentence similarity tasks.  However, it doesn’t have a significant impact on single sentence classification tasks, including sentiment analysis task and question-type classification.



2/ On combining both generative and discriminative objectives into a single multi-view framework. (suggested by Reviewer 2)

Models with both generative and discriminative objectives are trained to see if further improvement can be provided by combining an RNN encoder, an inverse of a linear decoder in the generative objective and a linear encoder in the discriminative objective.  The results of models trained on BookCorpus and UMBC News are presented in Table 3 in the appendix.

As presented in the table, no further improvement against models with only one objective is shown. In our understanding, the inverse of the linear decoder in generative objective behaves similarly to the linear encoder in the discriminative objective, which is presented in Table 6 in the main paper. Therefore, combining two objectives doesn’t perform better than only one of them.



3/ We will update our paper later to address more detailed issues mentioned by all three reviewers.



Thanks,

---

### Author Response · Authors · 2018-11-26
**Revision 2 is availble now**

We revised our paper slightly according to the comments from three reviewers, and we summarise the changes here:


1/ We elaborated the second contribution in our paper and discussed it in our paper.

Provided the results of models with generative and discriminative objective in Table 6, we confidently show that, in our multi-view frameworks with f and g, the two encoding functions improve each other’s view. By aligning representations produced from two distinctive information processing views, f gets improved on unsupervised tasks, and g gets helped on supervised ones.



2/ In general, aligning the representations generated from two distinct encoding functions ensures that the ensemble of them performs better.

On unsupervised evaluation tasks, simply averaging representations from two views without aligning them during learning leads to poor performance and it is worse than g (linear) encoding function solely.  Our multi-view frameworks ensure that the ensemble of two views provides better performance on both supervised and unsupervised evaluation tasks.




3/ The difference between our work and previous multi-view learning is stated more clearly in the introduction.

Compared to earlier work on multi-view learning (de Sa, 1993; Dhillon et al., 2011; Wang et al., 2015) that takes data from various sources or splits data into disjoint populations, our framework processes the exact same data in two distinctive ways.



--------------------------------(the following revisions come from Revision 1)-------------------------------


4/ The effect of the post-processing step. (suggested by Reviewer 1)

Six models trained on BookCorpus, UMBC news and Amazon Book Review with generative or discriminative objective are evaluated without the post-processing step that removes the first principal component. The results are presented in Table 2 in the appendix.

Overall, the postprocessing step overall improves the performance of our models on unsupervised evaluation tasks, and also improves the models with generative objective on supervised sentence similarity tasks.  However, it doesn’t have a significant impact on single sentence classification tasks, including sentiment analysis task and question-type classification.



5/ On combining both generative and discriminative objectives into a single multi-view framework. (suggested by Reviewer 2)

Models with both generative and discriminative objectives are trained to see if further improvement can be provided by combining an RNN encoder, an inverse of a linear decoder in the generative objective and a linear encoder in the discriminative objective.  The results of models trained on BookCorpus and UMBC News are presented in Table 3 in the appendix.

As presented in the table, no further improvement against models with only one objective is shown. In our understanding, the inverse of the linear decoder in generative objective behaves similarly to the linear encoder in the discriminative objective, which is presented in Table 6 in the main paper. Therefore, combining two objectives doesn’t perform better than only one of them.



Minor issues are also addressed in this revision to make the paper clearer.

Thanks,

---

### Meta-Review · Area_Chair1 · 2018-12-13
**Small but reasonable novel contribution**

**Confidence:** 2
**Recommendation:** Reject

**Metareview:**

This paper offers a new method for sentence representation learning, fitting loosely into the multi-view learning framework, with fairly strong results. The paper is clearly borderline, with one reviewer arguing for acceptance and another arguing for rejection. While it is a tough decision, I have to argue for rejection in this case.

There was a robust discussion and the authors revised the paper, so none of the remaining technical issues strike me as fatal. My primary concern is simply that the reviewers could not reach a consensus in favor of the paper. In particular, two reviewers expressed concerns that this paper makes too small an advance in NLP to be of interest to non-NLP researchers. I think it should be possible to broaden the scope of the paper and resubmit it to another general ML venue, and (as one reviewer suggested explicitly), this paper may have a better chance at an NLP-specific venue.

While neither of these factors was crucial in the decision, I'd encourage the authors (i) to put more effort into comparing properly with the Subramanian and Radford baselines, and (ii) to clarify the points about the human brain. For the second point: While none of the claims about the brain are false *or misleading*, as far as I know, the authors do not make a convincing case that the claims about the brain are actually relevant to the work being done here.